# Steroidal Saponins: Naturally Occurring Compounds as Inhibitors of the Hallmarks of Cancer

**DOI:** 10.3390/cancers15153900

**Published:** 2023-07-31

**Authors:** Salwa Bouabdallah, Amna Al-Maktoum, Amr Amin

**Affiliations:** 1Theranostic Biomarkers, LR23ES02, Faculty of Medicine of Tunis, Université Tunis El Manar, Tunis 1006, Tunisia; 2Biology Department, College of Science, United Arab Emirates University, Al Ain 15551, United Arab Emirates; amnamalmaktoum@gmail.com

**Keywords:** steroidal saponins, cancer, natural products, hallmarks of cancer, cancer treatment, mechanism of cancer

## Abstract

**Simple Summary:**

Cancer is one of the most life-threatening diseases that affects a rapidly growing number of individuals worldwide. Currently, chemotherapy is the most common method of treating cancer. The unfavorable results involved with existing cancer treatments have motivated the search for novel approaches to treat cancer. Many of these treatments exert their anti-cancer effects by targeting the hallmarks of cancer. The hallmarks of cancer are a set of traits shared among almost all cancers. The increasing knowledge of these characteristics has provided a framework for the development of promising anti-cancer drugs. Historically, nature has been a valuable source for the discovery of effective anti-cancer agents. Steroidal saponins are a group of naturally occurring compounds that have been recently explored for their anti-cancer properties. This review summarizes key findings from recently published articles on the use of steroidal saponins as inhibitors of the hallmarks of cancer.

**Abstract:**

Cancer is a global health burden responsible for an exponentially growing number of incidences and mortalities, regardless of the significant advances in its treatment. The identification of the hallmarks of cancer is a major milestone in understanding the mechanisms that drive cancer initiation, development, and progression. In the past, the hallmarks of cancer have been targeted to effectively treat various types of cancers. These conventional cancer drugs have shown significant therapeutic efficacy but continue to impose unfavorable side effects on patients. Naturally derived compounds are being tested in the search for alternative anti-cancer drugs. Steroidal saponins are a group of naturally occurring compounds that primarily exist as secondary metabolites in plant species. Recent studies have suggested that steroidal saponins possess significant anti-cancer capabilities. This review aims to summarize the recent findings on steroidal saponins as inhibitors of the hallmarks of cancer and covers key studies published between the years 2014 and 2024. It is reported that steroidal saponins effectively inhibit the hallmarks of cancer, but poor bioavailability and insufficient preclinical studies limit their utilization.

## 1. Introduction

Cancer prevails as a global clinical concern accounting for approximately 19 million cases and 10 million deaths in 2020 [1]. It has been estimated that these dire figures will continue to rise regardless of the rapid advances in cancer management and treatment [1]. Cancer is characterized by a disruption of cellular homeostasis resulting in unconstrained cell division. Underlying mechanisms contributing to cancer development involve modifications in signaling pathways that confer proliferative, angiogenic, metastatic, and anti-apoptotic capabilities in tumor cells. Such characteristics have been thoroughly elucidated by Hanahan and Weinberg as the “Hallmarks of Cancer”. Initially, only six hallmarks were proposed in 2000. Upon further research, the list has been updated in 2011, and again in 2022, with a total of fifteen hallmarks [2,3,4]. Despite the heterogeneity of cancer, these hallmarks are shared among almost all cancers. The hallmarks of cancer have consequently been targeted in the development of therapeutic drugs, including various chemotherapies (Figure 1). Chemotherapy is the most common conventional cancer treatment and is often employed in combination with surgery and radiotherapy [5]. Many cancers are inoperable and gain resistance against chemotherapeutic drugs [6]. Intolerable drug-induced side effects may also arise, leading to a reduction in or discontinuation of the treatment [7,8]. The challenges pertaining to such therapies have motivated the search for alternative approaches to cancer treatment that are effective but less invasive at the same time.

Nature has offered a reliable reservoir for the discovery of novel pharmacological agents [9]. Recently, a growing body of research has suggested that natural biomolecules, particularly phytochemicals, could interrupt the initiation, development, and progression of various cancers [10,11], including alkaloids, flavonoids, and saponins [12,13]. In fact, almost half of all approved anti-cancer drugs from the years 1981 to 2019 are of natural origin [14]. Therefore, it is within reason to resort to natural ingredients in the hunt for improved cancer regimens.

Saponins are a diverse class of naturally occurring compounds that are primarily distributed among plant species existing as secondary metabolites. Structurally, they are made up of a hydrophobic backbone that is linked to a hydrophilic glycan consisting of polysaccharide chain(s). The composition of the aglycone backbone, often referred to as sapogenin, determines the classification of saponins into two major groups. Respectively, steroidal or triterpenoid saponins have a backbone made of steroid or triterpenoid aglycone (Figure 2) [15]. This structural variability of saponins allows diversity in their biological activities [16]. Steroidal saponins have been shown to exert antimicrobial, anti-inflammatory, anti-diabetic, and anti-cancer activities [17,18,19]. They have been particularly gaining attention for their ability to inhibit various cancers both in vitro and in vivo (Table 1 and Table 2). This review aims to summarize the recent findings on the anti-cancer potential of steroidal saponins in the context of their effects on the hallmarks of cancer in the hopes of providing an updated and comprehensive foundation for further research. These hallmarks include sustaining proliferative signaling, evading growth suppressors, avoiding immune destruction, enabling replicative immortality, tumor-promoting inflammation, activating invasion and metastasis, inducing angiogenesis, genome instability and mutation, resisting cell death, and deregulating cellular energy. The studies covered in this review include those published between the timeframe of January 2014 and January 2024. Keywords including “steroidal saponins”, “cancer”, “cell proliferation”, “inflammation”, “metastasis”, “apoptosis”, and “angiogenesis” were used in PubMed, ScienceDirect, and Google Scholar to source primary research articles. Due to the overwhelming number of articles published within this period, the articles were exclusively selected based on the aforementioned keywords and the article’s relevance to the hallmarks of cancer.

## 2. Inhibition of Proliferative Signaling of Cancer Cells

Perhaps the most well-known feature of cancer cells is their unrestrained proliferation. The production and release of signals that promote cell growth and proliferation are highly regulated in normal cells to ensure that cellular equilibrium is maintained. Cancer cells, however, have the ability to sustain proliferative signaling and divide uncontrollably. This trait is acquired by cancer cells via autocrine signaling, the stimulation of neighboring normal cells to supply growth factors, the loss of negative regulators, and growth factor receptor overexpression and hypersensitivity [2].

Transmembrane growth factor receptors are responsible for the initiation of downstream signaling cascades involved in cell proliferation and are overtly displayed on the surface of cancer cells. Targeting these growth receptors is considered a promising strategy in cancer treatment as exemplified by established cancer drugs such as gefitinib, lapatinib, and trastuzumab [69]. A steroidal saponin obtained from the rhizomes of *Paris polyphylla* var. *latifolia* was shown to target epidermal growth factor receptor (EGFR) signaling in the glioma cell lines LN229 and U251, as indicated in a network pharmacology analysis. The key proteins of proliferative signaling such as EGFR, PI3K, Akt, and mTOR in their phosphorylated forms have been downregulated as a result of steroidal saponin treatment in the cell lines [37]. *Paris polypylla* steroidal saponins diosgenin, pennogenin, and 7-ketodiosgenin acetate were shown to act in a similar manner against breast cancer receptors, EGFR, and estrogen receptor-α (Erα). Molecular docking studies suggest that the binding affinity of the three compounds to the aforementioned receptors is higher than the binding of tyrosine kinase inhibitor erlotinib [30]. In another in silico study, diosgenin and monohydroxy spirostanol derived from *Prunus dulcis* seeds exhibited binding affinities to EGFR and Human Epidermal Growth Factor Receptor 2 (HER2) comparable to cancer drugs such as tak-285 and lapatinib [31].

The activation of the transcription factor STAT3 is observed in many cancers contributing to their sustained proliferative signaling [70,71]. The steroidal saponin diosgenin possessed inhibitory capabilities against STAT3 in the colon cancer cell line SW480 according to molecular docking and Western blot analyses. EdU (5-ethynyl-2′-deoxyuridine) staining and cell colony formation tests suggested that diosgenin caused an inhibition of cell proliferation in a dose-dependent manner and exceeded the inhibitory effects of the reference drug cisplatin. These results were further verified in a colon cancer nude mouse model [32]. Similarly, diosgenin downregulated the activation of the NF-κB/STAT3 pathway in the transgenic prostate cancer mouse model by preventing the phosphorylation of IKK and subsequently reducing IκBα and p65 phosphorylation [33].

Multiple studies investigating the effects of three steroidal saponins isolated from *Aspidistra letreae* have reported anti-cancer properties against a wide spectrum of tumor cells in vitro. Aspiletreins A, B, and C have been tested against H460, H23, A549, LU-1, HeLa, MDA-MB-231, HepG2, and MKN-7 human cancer cell lines, providing promising anti-proliferative effects that were evaluated by SRB and MTT assays [22,24]. Iksen et al. (2023) investigated the molecular mechanisms by which these steroidal saponins exert their anti-cancer properties on NSCLC using in silico approaches such as molecular docking and network pharmacology analysis [23]. The findings indicated that Aspiletreins A, B, and C primarily target STAT3 in its phosphorylated state, which is commonly observed in lung adenocarcinomas. To validate these results, an in vitro experiment was conducted using Aspiletrein B against the NSCLC cell line H460. Aspiletrein B was selected since it was the most potent compound out of the three steroidal saponins according to MTT assay results. The suggested molecular target of Aspiletrein B was substantiated by a notable decrease in the active phosphorylated STAT3, while its inactivated form remained unaffected [23].

## 3. Cancer Cell Death Induction

Cell death evasion is a classical hallmark of cancer [2]. Recently, a large body of literature has linked the anti-cancer properties of steroidal saponins with their ability to instigate tumor cell death. In the case of steroidal saponins, cell death is mediated by apoptotic, autophagic, ferroptotic, and necroptotic mechanisms (Figure 3).

### 3.1. Apoptosis

Apoptosis, a form of programmed cell death, is a tightly regulated physiological process during which a series of biochemical events cause a dismantling of unnecessary cells to maintain tissue homeostasis. This process has been of significant relevance in cancer treatment since it can eliminate cancer cells without eliciting an inflammatory reaction.

PP9, a steroidal saponin obtained from *Paris polyphylla*, was shown to selectively induce the apoptosis of HT-29 and HCT116 cell lines with no significant cytotoxicity observed in normal human colon epithelial cells. This was evident in a notable decrease in anti-apoptotic Bcl-2 levels, while pro-apoptotic proteins Bax, cleaved caspase 9, cleaved caspase 3, and cleaved PARP increased, indicating the intrinsic activation of apoptosis. Immunoblot analyses suggested that PP9 significantly suppressed the PI3K/Akt/GSK3β signaling pathway, which in turn induced cell cycle arrest and ultimately apoptosis. This cytotoxic effect was further verified in an HCT116 xenograft mouse model, without inducing overt toxicity in non-cancerous cells. This steroidal saponin outperformed the anti-cancer effects of 5-Fu in vivo [42].

Trilliumosides A and B are steroidal saponins derived from the rhizomes of *Trillium govanianum*. A recent study has explored their cytotoxic potential against a wide range of cancer cell lines and found the most promising results in A-549 and SW-620 cell lines. Apoptotic morphological features such as chromatin shrinkage and nuclear condensation have been observed in a dose-dependent manner in A-549 cells treated with Trilliumoside B. Since the intracellular accumulation of ROS is indicative of apoptosis, fluorescent product dichlorofluorescein (DCF) staining was conducted to assess ROS levels. This confirmed increased levels of ROS in A-549 cells treated with Trilliumoside B. Trilliumoside B was also shown to decrease the mitochondrial membrane potential, which is associated with apoptosis. It reduced antiapoptotic protein Bcl-2 levels and activated Bax and caspase 3 [48].

The steroidal saponin protodioscin induced apoptosis in bladder cancer cell lines 5637 and T24 cells. Transcriptome and Western blot analyses were performed to identify the mechanisms by which protodioscin exerts its cytotoxic effects. The results indicated that protodioscin modulates the PI3K/Akt/mTOR signaling pathway. It also activated p38 and JNK signaling pathways, which resulted in ER-stress-dependent apoptosis. The results were further verified in a nude mouse xenograft model in which tumor growth was substantially inhibited [43].

Bufalin, a steroidal saponin obtained from Chinese toad venom, induced apoptosis in the glioma cell line U251. It caused the inhibition of mitochondrial Na^+^/K^+^ATPase activity and an overload of Ca^2+^ in the mitochondria. This mitochondrial dysfunction led to the generation of ROS and the release of cytochrome C and cleaved caspase 3 [27]. In another study, the anti-cancer effects of bufalin on glioblastoma were evaluated in vivo and in vitro. Similarly, bufalin induced apoptosis by decreasing the mitochondrial membrane potential and increasing mitochondrial permeability, leading to the release of cytochrome c and caspase activation [26].

The steroidal saponin pennogenin-3-α-L-rhamnopyranosyl-(1→4)-[α-Lrhamno-pyranosyl-(1→2)]-β-D-glucopyranoside (N45) was shown to possess pro-apoptotic activities glioblastoma cell lines U87 and U251. This was substantiated by the TUNEL assay, flow cytometry, and Western blot analysis. N45 also sensitized drug-resistant U87 cells (U87R) to the anti-cancer drug temozolomide (TMZ) via apoptosis induction. The steroidal saponin decreased the levels of the drug-resistant biomarker MGMT in U87R cells. The upregulation of pro-apoptotic proteins including cytochrome C, BAX, and cleaved caspase 3 was detected in treated U87R cells. This intrinsic pathway of apoptosis was activated by a downregulation of the PI3K/Akt signaling pathway due to intracellular ROS accumulation [61].

The total steroidal saponins derived from the roots and rhizomes of *Trillium tschonoskii* exhibited a protective advantage in a 1, 2-dimethyl-hydrazine (DMH) and dextran sodium sulfate (DSS)-induced colorectal cancer mouse model. The tumor size in the animals significantly decreased following *Trillium tschonoskii* steroidal saponins (TTS) administration in a dose-dependent manner. This inhibition of tumor growth may be attributable to TTS-mediated apoptosis, according to TUNEL and caspase-3 activity assay results. The apoptotic effects of TTS were further verified in HT-29 cells. TTS demonstrated growth inhibitory efficacy, which is comparable to the anti-cancer drug 5-Fu in HT-29 cells. TTS caused apoptosis by inhibiting the activation of ERK1/2, JNK, and p38 signaling [72].

Paris saponins (PS) I, II, VI, and VII induced apoptosis in the gefitinib-resistant non-small cell lung cancer cell line PC-9-ZD, as validated by flow cytometry, Hoechst staining, and Western blot results. The combination of each of the tested Paris saponins with gefitinib generated a significantly higher apoptotic rate when compared to gefitinib or PS monotherapies. PSVII and PSI had the most significant pro-apoptotic effects in this study. The Western blot analysis indicated that the inactivation of the PI3K/Akt pathway contributed to apoptosis induction [73].

The steroidal saponin cholestanol glucoside (CG) isolated from *Lasiodiplodia theobromae* presented a considerable degree of cytotoxicity against A549, PC3, and HEPG2 cells with A549. A549 were the most susceptible to the treatment and were, therefore, selected for further investigation. Steroidal saponin treatment increased intracellular ROS levels and mitochondrial membrane permeability in the A549 cells. These cellular events ultimately led to the apoptosis of the lung cancer cells [52].

Polyphyllin VII is a steroidal saponin isolated from Paris polyphylla and has been shown to induce apoptosis in the lung cancer cell line A549 via the inhibition of PI3K/Akt and NF-κB signaling. The downregulation of these proteins led to a decrease in mitochondrial membrane potential and ultimately the induction of apoptosis. The PI3K inhibitor wortmannin was used in combination with polyphyllin VII, extenuating the compound’s inhibitory effects on NF-κB p65, and consequently increasing the rate of apoptosis in A549 cells [53].

Oleandrin, a steroidal saponin found in the seeds and leaves of *Nerium oleander*, demonstrated selective cytotoxic properties against breast cancer cell lines MDA-MB-231, MCF7, and SK-BR-3. Oleandrin treatment did not significantly affect the mammary epithelial cell line MCF0A. Out of the tested cell lines, SK-BR-3 exhibited the highest susceptibility to oleandrin. Cells with distinct apoptotic morphological characteristics, such as nuclear pyknosis and fragmentation, increased following the oleandrin treatment. Oleandrin caused an increased expression of pro-apoptotic proteins bax and bim while decreasing anti-apoptotic bcl-2 expression. This intrinsic apoptotic pathway was activated by an oleandrin-induced ER stress response. This was corroborated by the Western blot results in which the expression of phospho-PERK and its downstream proteins phospho-eIF2α, ATF4, and CHOP increased after oleandrin treatment [63]. Oleandrin also had pro-apoptotic effects against several other cell lines, including SW480, HCT116, RKO, A375, GL261, and U87MG cells [62,64]. It was reported that DNA damage induction contributes to the pro-apoptotic effects of oleandrin. This was shown in the downregulation of RAD51 and the upregulation of XRCC1 and γH2AX in A549 cells after oleandrin treatment [65].

### 3.2. Autophagy

Autophagy is a cellular mechanism in which worn-out cellular components are degraded or recycled to meet the requirements of starving cells. In the context of cancer treatment, autophagy has a paradoxical role in that it can either promote tumor cell survival or induce autophagic cell death. Autophagic cell death is another form of programmed cell death involving the excessive degradation of vital cellular components, thereby halting cell survival. While some inhibitors of autophagy have been proposed to augment the effects of chemotherapy [74], other anti-cancer agents, including steroidal saponins, trigger autophagic cancer cell death [75].

The steroidal saponin A-24, collected from *Allium chinense*, induced autophagy and apoptosis in SGC-7901, AGS, and KATO-III gastric cancer cell lines. In SGC-7901 and AGS cells, the induction of autophagy was verified by p62 degradation and the upregulation of LC3-II, Beclin-1, and GFP-LC3. To assess the molecular mechanisms of this cytotoxicity, the proteins of the PI3K/Akt/mTOR pathway were quantified. The mTOR protein is made up of two protein complexes, one of which is involved in inhibiting the initiation of autophagy. A downregulation of Akt and mTOR in their phosphorylated forms was observed following the treatment of SGC-7901 and AGS cells with A-24, indicating the induction of autophagy [20]. The effects of A-24 on p53-null and p53-wild type KATO-III cells indicated that ROS-accumulation-mediated apoptosis was independent of p53. A-24-induced autophagy was negatively influenced by p53; however, it still occurred regardless of p53 status [21].

*Paris polyphylla* is a plant species known for its abundance of steroidal saponins. The gross steroidal saponin yield from the root and rhizome of Paris polyphylla was shown to induce autophagy and apoptosis in A549 cells in a time-dependent manner. Paris polyphylla steroidal saponins (PPSS) increased the levels of the autophagic markers LC3-II and Beclin 1 in the cancer cells [76]. Paris Saponin VII (PSVII) restricted the proliferation of breast cancer cell lines MDA-MB-231, MDA-MB-436, and MCF-7 by eliciting autophagic cell death and caspase-dependent apoptosis [41]. The steroidal saponin substantially reduced the activity of the Yes-associated protein (YAP). This protein is a downstream effector in the Hippo signaling pathway and is involved in the transcription of proliferative genes. The Hippo-YAP pathway is considered a regulator of autophagy and the inhibition of YAP has been linked to the induction of autophagic cell death of tumor cells [75]. Molecular docking results revealed that PSVII binds to the MST2-MOB1-LATS1 complex, which activates LATS1. LATS1 is an enzyme that inactivates YAP, limiting its nuclear translocation and anti-autophagic transcriptional activities. When PSVII was used to treat an MDA-MB-231 xenograft mouse model, LC3-II, Beclin 1, and the YAP-inhibitor LATS1 were upregulated. YAP, along with the proliferative marker Ki67, were downregulated in the tumor tissue [41].

The total steroidal saponins occurring in *Solanum americanum* were shown to overcome multidrug resistance (MDR) in the adriamycin-resistant leukemia cell line K562/ADR by eliciting autophagic cell death. SN treatment resulted in caspase-dependent apoptosis in the cell lines as indicated in flow cytometry and Western blot analyses. However, the use of a pan-caspase inhibitor suggested that apoptosis is not the primary mechanism by which SN reverses MDR. The levels of autophagy-associated proteins were assessed using Western blotting post-treatment. SN caused a notable rise in Beclin-1 and LC3-II and a reduction in p62/SQSTM1. The induction of autophagy was further verified using the autophagy blockers 3-methyladenine, Chloroquine, or ATG5 silencing. These inhibitors increased cancer cell viability and upregulated drug-resistant protein expression, thereby indicating that autophagy has a major role in reversing drug resistance in K562/ADR cells. The findings of this study were verified in a K562/ADR xenograft model [77]. The steroidal saponin S-20 is an important constituent of SN. It demonstrated similar cytotoxic effects against adriamycin-sensitive and adriamycin-resistant leukemia cell lines K562 and K562/ADR via autophagic cell death [54].

### 3.3. Ferroptosis

Ferroptosis is a relatively newly discovered iron-dependent mechanism of cell death that is associated with intracellular lipid peroxide accumulation [78]. Steroidal saponins such as dioscin, schizocapsa plantaginea hance I (SSPH I), timosaponin AIII, and polyphyllin B and III evoked ferroptosis in different cancer cells. SSPH I was found to induce ferroptosis in HepG2 liver cancer cells. This was confirmed by an increase in ROS, malondialdehyde, transferrin, and ferroportin proteins and a decrease in glutathione levels following the SSPH I treatment [44]. Dioscin had cytotoxic effects against melanoma cell lines A375, G361, and WM115 via ferroptosis induction. This was verified by a surge in intracellular iron levels via the modulation of transferrin and ferroportin levels. Dioscin was potent alone but also enhanced the anti-cancer effects of rapamycin, cisplatin, dacarbazine, and vemurafenib when used in combination [28]. Timosaponin AIII caused ferroptotic cell death in non-small cell lung cancer (NSCLC) cells both in vivo and in vitro. It acts by binding to HSP90, which consequently leads to the degradation of glutathione peroxidase 4 (GPX4)—a ferroptosis inhibitor. Around 90% of all NSCLC cells were eliminated via ferroptosis in this investigation [46]. Polyphyllin B was shown to be mechanistically similar to dioscin when examined against gastric cancer in vitro. It induced ferroptosis by directly binding to GPX4. This was confirmed by molecular docking, Western blot, and immunofluorescence analyses [55]. Another *Paris polyphylla*-derived steroidal saponin, polyphyllin III, induced the ferroptotic death of MDA-MB-231 triple-negative cancer cells via the ACSL4-dependant peroxidation of lipids. However, polyphyllin III also increased the levels of the anti-ferroptotic protein xCT, thereby reducing the potency of the compound. Combining polyphyllin III with a xCT inhibitor enhanced the anti-cancer efficacy of polyphyllin III. This combination therapy exhibited notable tumor-suppressive effects in an in vivo xenograft model [56].

### 3.4. Necroptosis

The pro-apoptotic activities of polyphyllin D, a major constituent of *Paris polyphylla*, were examined on the human neuroblastoma cell lines IMR-32, LA-N-2, and NB-69. Cell viability analysis revealed that NB-69 was the most susceptible to the anti-cancer effects of polyphyllin D. The activation of caspases 3, 7, and 8 was also observed in the treated NB-69 cells but not in IMR-32 and LA-N-2 cells. These findings suggest that polyphyllin D induces an alternative mechanism of cell death to restrict the viability of the neuroblastoma cell lines. The necroptosis inhibitors necrostatin-1 and necro sulfonamide were utilized to assess the induction of necroptosis. The treatment of the neuroblastoma cell lines with necroptosis inhibitors caused a significant reduction in polyphyllin D-mediated cell death. These results indicate that necroptosis may be associated with the anti-cancer activities of polyphyllin D [57].

## 4. Inhibition of Replicative Immortality

Telomeres are repetitive regions of DNA existing at the ends of chromosomes. Normally, telomeres decrease in length with every successive round of replication. This phenomenon contributes to a cell’s mortality as the erosion of telomeric DNA leads to cellular senescence and ultimately cell death. Cancer cells circumvent this limited replicative potential and become immortal with an unlimited life span. They attain this capability primarily by overexpressing telomerase to ensure extended telomeres [2].

Diosgenin was shown to inhibit the expression of the telomerase reverse transcriptase (TERT) gene of rat C6 and human T98G glioblastoma cell lines. This gene encodes for an essential subunit of telomerase, and its inhibition prevents the maintenance of extended telomeres in cancer cells [34]. Diosgenin also inhibited the human telomerase reverse transcriptase (hTERT) gene expression in A549 cells verified by qRT-PCR analysis. Pure diosgenin had a greater impact on hTERT inhibition as opposed to fenugreek extract diosgenin [36].

## 5. Inhibition of Tumor Promoting Inflammation

Inflammatory conditions have been long known to be a breeding ground for the development and progression of various tumors. In the setting of chronic inflammation, growth factors and cytokines are released from immune cells into the tumor microenvironment, thereby promoting the growth of tumors [2]. Targeting this tumor-promoting inflammation is a major therapeutic intervention for the treatment of cancer. Studies have indicated that steroidal saponins have the ability to suppress tumor-promoting inflammation by inhibiting inflammatory signaling pathways.

The steroidal saponin Diosgenin decreased inflammation in the NSCLC cell line A549 by modulating NF-kB signaling [35]. The transcription factor NF-kB is essential for the transcription of pro-inflammatory genes such as those encoding for COX-2 and is overexpressed in numerous cancers [79]. The detection of NF-kB via fluorescence immunocytochemistry indicated that diosgenin inhibits the nuclear import of NF-kB and consequently prevents the expression of pro-inflammatory genes. This was verified by a reduction in COX-2 and PGE2 expression as a result of diosgenin treatment according to Western blot analysis [35].

Tumor-associated macrophages (TAM) facilitate the growth and proliferation of tumors and have a critical role in the tumor microenvironment. Dioscin caused a phenotypic shift of pro-tumorigenic M2-like TAMs into the tumor-suppressing M1-like TAMs in the Raw264.7 cell line. It also limited the secretion of cytokines such as IL-10 and induced the phagocytosis BMDMs. These results were further verified in a subcutaneous lung tumor animal model [80].

RCE-4 obtained from *Reineckia carnea* exerted anti-inflammatory activities in a human cervical cancer Caski cell xenograft mouse model. Paclitaxel was used as a reference drug in this investigation. RCE-4 caused a reduction in the COX-2 expression, which was assessed via immunohistochemical analysis. This effect was comparable to that of paclitaxel but did not outperform palitaxel’s overall tumor-suppressing activities in the animal model [68].

## 6. Inhibition of Tumor Invasion and Metastasis

A common feature shared among malignancies is their ability to spread throughout an organism and colonize different sites. Cancer cells are subjected to various molecular and genetic modifications that allow them to migrate and conform to new distant microenvironments [2]. Considering that the metastatic potential of tumors is a major contributor to the lethality of cancer, it is important that this feature is targeted when developing cancer treatments [81]. Steroidal saponins have been shown to alleviate this hallmark of cancer by inhibiting key players in tumor metastasis and invasion. This includes Epithelial-mesenchymal transition (EMT) proteins and matrix metalloproteinases (MMPs).

The steroidal saponin protodioscin mitigates 5637 and T24 bladder cancer cell migration and invasion. This was assessed using a wound-healing assay, motility assay, and a transwell Matrigel invasion assay. Since EMT is linked to the metastatic abilities of cancer cells, associated proteins were evaluated in a Western blot. The results indicated a downregulation in N-cadherin and an upregulation in E-cadherin protein expression, confirming the inhibition of tumor metastasis [43].

The steroidal saponin A-24 inhibited the migration of gastric cancer cell lines. p53-deficient KATO-III cells and p53 wild-type gastric cancer cells were utilized in this study to determine the role of p53 in cancer cell migration and invasion. The downregulation of MMP-2 expression and a reduction in transwell migration and wound closure rates were observed in a dose-dependent manner. These results were independent of p53 status [21].

Bufalin prevented cancer cell invasion and migration in the gallbladder cancer cell line GBC-SD according to transwell migration and invasion results. It also lowered the expression of Snail and MMP9 while increasing the E-cadherin levels. Since cancer stem cells promote metastasis, stemness marker proteins CD133, CD44, Sox2, Oct4, and Nanog were assessed. The expression of these proteins was almost diminished after bufalin treatment [25].

Trillin derived from *Trillium tschonoskii* prohibited the migration and invasion of the liver cancer cell lines HepG2 and PLC/PRF5, as indicated in a wound healing assay. Genes involved in cell migration and invasion were suppressed after trillin treatment, as confirmed by Western blot and qPCR results. A significant reduction in the expression of STAT3, MMP1, MMP2, MucI, and VEGF was observed. This study suggested that trillin mainly targets STAT3, which is responsible for the expression of genes encoding for proteins with significant roles in the migration and invasion of tumor cells [47].

The total steroidal saponins from *Paris polyphylla* (PPSS) were shown to limit the migration and invasion of A549 cells. This was evaluated by Matrigel invasion chamber and wound-healing assays. PPSS suppressed the expression and activity of the metalloproteinases MMP-2 and MMP-9, as indicated by Western blot and gelatin zymography results [82]. Major constituents of PPSS included polyphyllin I, polyphyllin II, and polyphyllin E.

Polyphyllin I restricted the proliferation and invasion of the cisplatin-resistant gastric cancer cell line SGC7901/DDP via the modulation of the CIP2A/PP2A/Akt pathway. CIP2A plays an essential role in tumor metastasis and EMT and its upregulation is often observed in gastric cancer. Polyphyllin I also decreased tumor volume and CIP2A, phospho-Akt, and vimentin expression in a xenograft murine model [58]. These results were consistent with the treatment of prostate cancer with polyphyllin I in vitro and in vivo [60]. It was also reported that polyphyllin I inhibited the migration of the osteosarcoma cancer cell lines 143-B and HOS in a wound-healing assay and a xCELLigence RTCA DP system. This inactivation of EMT is ascribed to the polyphyllin I-mediated inhibition of Wnt/β-catenin signaling [59].

Polyphyllin II suppressed tumor cell motility, as observed in wound-healing assays. It also reduced cofilin activity and MMP2, and MMP9 expression in the liver cancer cell lines HepG2 and BEL7402. This inhibition of tumor cell motility was achieved as polyphyllin II targets Akt/NF-kB signaling [49]. These effects were comparable to the treatment of ovarian cancer cell lines SK-OV-3 and OVCAR-3 with polyphyllin E. MMP2 and MMP9 were downregulated primarily via the inhibition of the Akt/NF-kB pathway post-polyphyllin E treatment [51]. In another study, polyphyllin II inhibited the migration of T24 and 5637 bladder cancer cells in a wound-healing assay. It significantly decreased N-cadherin, MMP2, MMP9, and other EMT-associated proteins to a level resembling that of normal cells [50].

## 7. Inhibition of Abnormal Metabolism

Considering the rapid growth and proliferation of cancer cells, an increased demand for energy is required to sustain such processes. Therefore, cancer cells must rewire their metabolic programs to meet these high demands. However, this metabolic reprogramming of cancer cells poses a paradox since relatively inefficient glycolysis is the preferred mode of energy production for cancer cells, even under aerobic conditions. The term “aerobic glycolysis” is used to describe the glucose metabolism of cancer cells even in the presence of oxygen. This phenomenon is known as the Warburg effect [2]. It is important to note that in instances of glucose deprivation, cancer cells resort to mitochondrial metabolic pathways rather than aerobic glycolysis to meet their energy demands. The modulation of both of these metabolic pathways may be a more effective approach to terminating cancer cells [83].

The steroidal saponin gracillin was found to inhibit both aerobic glycolysis and OXPHOS in H460 and H226B NSCLC cells, thereby depriving tumor cells of all modes of ATP synthesis [38]. The anti-glycolytic effects of the steroidal saponin were confirmed by the modulation of metabolites involved in glycolysis, a decrease in lactate production, and the extracellular acidification rate following the gracillin treatment. The same results were observed in triple-negative breast cancer cell lines MDA-MB-468 and MDA-MB-231. The downregulation of the mitochondrial function and consequently OXPHOS was evaluated using the MTT assay. The best results were observed when gracillin was combined with the OXPHOS-mediated energy production inhibitors antimycin or oligomycin [38].

Dioscin demonstrated inhibitory effects on glycolysis in the colorectal cancer cell lines DLD1, HCT116, SW480, HT29, HCT8, and SW620 [29]. The protein Skp2 is overexpressed in colorectal cancers and is important for glycolysis to take place. It also promotes the expression of Glut1 transporters, which are essential for the increased influx of glucose into cancer cells to compensate for the low yield of ATP from aerobic glycolysis [84]. Dioscin markedly reduced glycolysis by attenuating phosphorylated Skp2 levels in the colorectal cancer cell lines. The effects of dioscin were further investigated in a xenograft model in which the expression of Skp2 was also reduced [29]. A notable characteristic of the Warburg effect is the cellular utilization of glutamine as an energy source and as a precursor for the synthesis of cellular components [85]. Dioscin was reported to interfere with D-glutamine metabolism in a SW480 rectal cancer cell line. Aerobic glycolysis is characterized by an increase in lactic acid production as pyruvic acids are converted into lactic acids. Dioscin also downregulated L-lactic acid levels in the same cancer cell line, suggesting the inhibition of aberrant pyruvate metabolism [86].

## 8. Targeting Cancer’s Evasion of Anti-Growth Signaling

In the absence of growth-stimulating signals, cells maintain dormancy via negative regulatory pathways that suppress cell proliferation and growth. This inhibition is primarily facilitated by tumor suppressor proteins and cell cycle checkpoints. Cancer cells bypass the inhibitory machinery that negatively regulates cell proliferation. Cyclins and cyclin-dependent kinases (Cdks) are essential for cell cycle progression and are often aberrantly expressed in cancer cells that escape growth-suppressing signaling.

A-24 arrested SGC-7901 and AGS gastric cancer cells at the G2/M phase of the cell cycle. CyclinB1 and cdc2 are important regulators of the cell cycle and allow cells to enter mitosis once they are activated. The activated protein kinase wee1 inhibits the activity of the cyclinB1/cdc2 complex, leading to G2/M phase arrest. A-24 caused a downregulation in cyclinB1 and activated cdc2 and upregulation in phosphorylated wee1. Xu et al. (2020) mentioned that these results are comparable to the steroidal saponin macrostemonoside A in the sense that both inhibit cancer cell growth via cell cycle arrest. However, A-24 had more promising anti-proliferative results when compared to macrostemonoside A [20].

The steroidal saponin extract obtained from the edible spears of wild asparagus was shown to induce G0/G1 cell cycle arrest in HCT116 cells. Treatment with the extract caused a reduction in cyclins D, A, and E according to Western blot results [87]. Similar results were observed when the total steroidal saponins from *Solanum americanum* were tested on K562 and K562/ADR cells. The treatment resulted in a G0/G1 cell cycle arrest with a reduction in cyclin E2 expression [77].

## 9. Inhibition of Angiogenesis

Angiogenesis is the formation of new blood vessels from pre-existing vasculature. In normal tissue, this progress occurs in a transient manner under specific physiological conditions. Contrarily, angiogenesis occurs perpetually in tumors to avoid cancer dormancy and to ensure that an adequate supply of oxygen and nutrients is continuously attained. Pro-angiogenic ligands such as vascular endothelial growth factors (VEGF) and fibroblast growth factors (FGF) are overly expressed in many cancers [2]. Currently, anti-angiogenic drugs such as sorafenib and pazopanib are in clinical use [88]. Therefore, the discovery of natural angiogenic inhibitors is a sound strategy for cancer treatment.

Paris saponins (PS) I, II, VI, and VII exhibited anti-angiogenic effects on HUVEC cells in a dose- and time-dependent pattern. According to MTT assays and tubule generation experiments testing the anti-cancer effects of the four steroidal saponins, PSI had the most potent inhibitory and anti-angiogenic effects. Therefore, PSI was selected for Western blot analysis. The results verified that PSI inhibited VEGFR2 phosphorylation in HUVEC cells. Additionally, the activation of proteins that are involved in the VEGFR2 pathway, including PI3K, Akt, mTOR, and S6K, was inhibited following the PSI treatment. PSI also activated p38, SRC, and eNOS, thereby changing vascular permeability and preventing angiogenesis [40].

The steroidal saponin terrestrosin D was tested on HUVECs and bladder-derived normal human microvascular endothelial cells to evaluate the anti-angiogenic capabilities of the compound. Terrestrosin D caused a halt in the growth of these cells in a dose-dependent manner. This inhibitory effect was further assessed in a PC-3 xenograft mouse model. Anti-CD31 antibody staining was employed to stain the tumor sections. The results indicated a notable inhibition of tumor angiogenesis in vivo following the terrestrosin D treatment. However, the steroidal saponin failed to reduce VEGF levels in the prostate cell line PC-3. Contrarily, the compound increased VEGF expression, which may be counteracted by anti-VEGF antibodies to extenuate the anti-cancer efficacy of terrestrosin D [67].

## 10. Anti-Tumor Immune Response Activation

A major hallmark of cancer is the ability of cancer cells to evade immune destruction. Tumor cells acquire this capability by downregulating antigens that are recognized by immune cells, making tumors undetectable under immune surveillance [2]. Steroidal saponins have been shown to induce an immune response against tumor cells that are otherwise undetectable. The steroidal saponin taccaoside A has demonstrated immunomodulatory effects in a T-cell and lung cancer H1299-GFP cell co-culture system. It regulated both CD4+ and CD8+ T-cells, leading to the lysis of tumor cells. Granzyme B (GZMB), an enzyme involved in the cytotoxic activities of T-cells, increased following the taccaoside A treatment. The same results were also observed for triple-negative breast cancer and melanoma, including anti-CTLA-4 therapy-resistant type melanoma. An in vivo mouse model confirmed the anti-tumor effects of taccaoside A, resulting in improved survival and a significant reduction in tumor size. It was also highlighted that taccaoside A activated the mTORC1/Blimp1 signaling pathway in T-cells, which enhanced their cytotoxic activities. This study suggests that taccaoside A could be a potential immunomodulatory anti-cancer agent, especially in patients that acquired resistance to other types of treatments [45].

## 11. Conclusions

The recent findings on the inhibitory effects of steroidal saponins against the hallmarks of cancer suggest that steroidal saponins can modulate cancer signaling pathways. Despite these promising findings, many challenges are yet to be overcome before the therapeutic use of steroidal saponins. The majority of the steroidal saponins that have been investigated within the past few years were tested on cancer cell lines. More in vivo studies are required to confirm the efficacy of steroidal saponins. To date, no clinical trials have assessed the safety and efficacy of steroidal saponins as cancer treatments in humans, which is needed to understand the full potential of steroidal saponins. An important impediment in preventing the progression of steroidal saponins into clinical stages may be associated with their increased toxicity. For instance, bufalin is known for its high toxicity, but efforts have been recently made to tackle this issue. A prodrug of bufalin, acetyl-buflain, has been shown to exert anti-cancer effects in vitro and in vivo with lower toxicity compared to bufalin [89].

Dioscin and diosgenin seem to be the most promising steroidal saponins due to their multifaceted anti-cancer activities that target most hallmarks of cancer. Both compounds surpassed the anti-cancer effectiveness of established reference drugs, such as adriamycin and cisplatin [32,90]. However, it has been reported that dioscin and diosgenin possess poor bioavailability, impeding their clinical development [91,92]. Efforts have been directed towards establishing structurally modified versions of the steroidal saponins to optimize their pharmacokinetic properties. For instance, the addition of a hydrophilic unit to diosgenin improved its bioavailability in vivo [93]. Furthermore, the use of novel drug delivery systems was shown to enhance diosgenin and dioscin pharmacokinetics in cancer therapy [94,95].

## Figures and Tables

**Figure 1 cancers-15-03900-f001:**
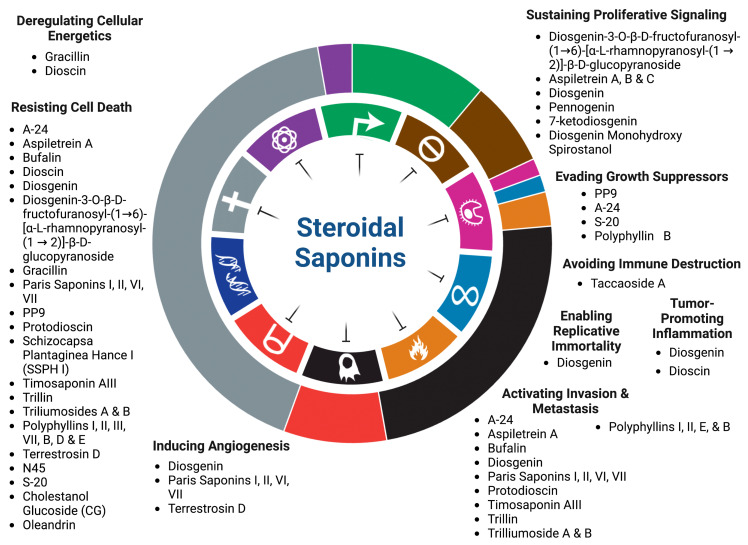
Steroidal saponins as inhibitors of the hallmarks of cancer.

**Figure 2 cancers-15-03900-f002:**
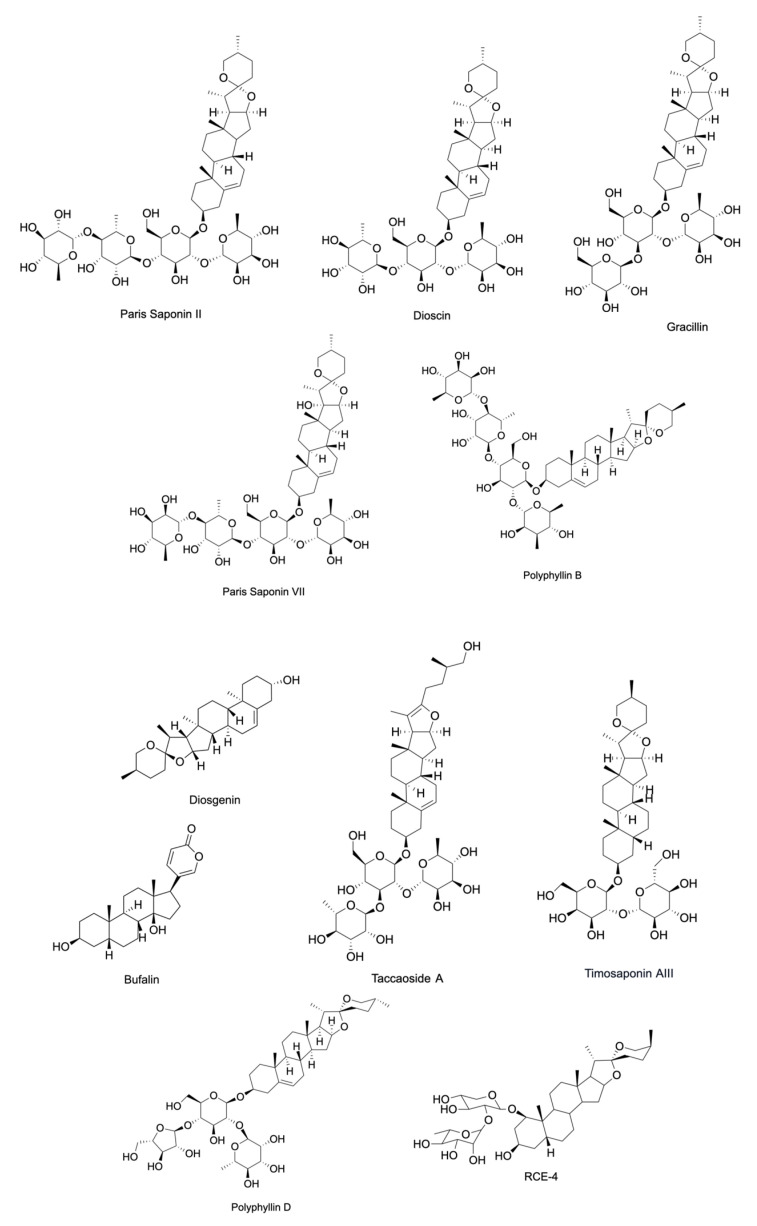
A representation of the chemical structure of steroidal saponins.

**Figure 3 cancers-15-03900-f003:**
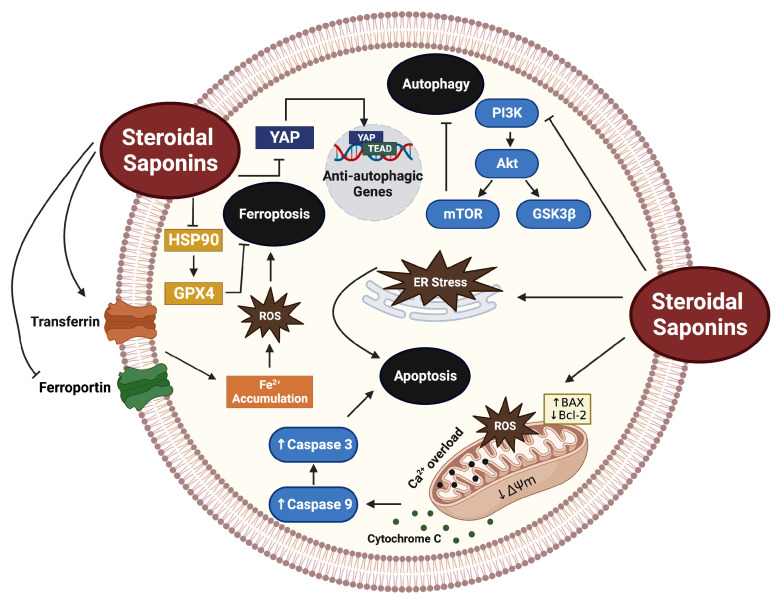
A schematic representation of the mechanisms by which steroidal saponins induce tumor cell death. The figure was constructed using BioRender.com.

**Table 1 cancers-15-03900-t001:** The anti-cancer effects of steroidal saponins in vitro. The half maximal inhibitory concentration (IC50) values are a reliable measure of the potency of steroidal saponins and their interference with cancer hallmark pathways. The IC50 values in Table 1 correspond to an exposure time of 24 h *, 48 h †, 72 h ‡ or mean ± SD.

Compound	Source	Cells	IC50	Targeted Hallmark of Cancer	References
A-24	*Allium chinense*	SGC-7901, AGS, MGC-803, NCI-N87, BGC-823, and KATO-III	3.03 * µM (SGC-7901), 2.18 * µM (AGS), 4.10 * µM (MGC-803), 4.53 * µM (NCI-N87), and 5.11 * µM (BGC-823).	Resisting Cell Death, Evading Growth Suppressors, Activating Tumor Invasion and Metastasis	[20,21]
Aspiletrein A	*Aspidistra letreae*	LU-1, HeLa, MDA-MB-231, HepG2, MKN-7, H460, H23 and A549	9.94 ‡ µM (LU-1), 7.69 ‡ µM (HeLa), 7.75 ‡ µM (MDA-MB-231), 9.19 ‡ µM (HepG2), 9.39 ‡ µM (MKN-7), 15.13 * µM (H460), 10.10 * µM (H23), and 8.78 * µM (A549)	Sustaining Proliferative Signaling, Activating Invasion and Metastasis, Resisting Cell Death	[22,23,24]
Aspiletrein B	*Aspidistra letreae*	LU-1, HeLa, MDA-MB-231, HepG2, MKN-7, and H460	20.27 ‡ µM (LU-1), 12.54 ‡ µM (HeLa), 20.46 ‡ µM (MDA-MB-231), 16.07 ‡ µM (HepG2), 18.15 ‡ µM (MKN-7), and 6.82 * µM (H460)	Sustaining Proliferative Signaling	[23,24]
Aspiletrein C	*Aspidistra letreae*	LU-1, HeLa, MDA-MB-231, HepG2, MKN-7, and H460	10.10 ‡ µM (LU-1), 9.03 ‡ µM (HeLa), 9.09 ‡ µM (MDA-MB-231), 8.84 ‡ µM (HepG2), 11.82 ‡ µM (MKN-7), and 15.75 * µM (H460)	Sustaining Proliferative Signaling	[23,24]
Bufalin	*Bufo gargarizans*	U251, SK-N-BE, SH-SY5Y, A549, GBC-SD	90 ‡ nM (SK-N-BE) and 30 ‡ nM (SH-SY5Y)	Resisting Cell Death, Activating Invasion and Metastasis	[25,26,27]
Dioscin	*Dioscorea zingiberensis* *and Dioscorea nipponica*	HUVEC, A375, G361, and WM115	-	Tumor Promoting Inflammation, Resisting Cell Death, and Deregulating Cellular Energetics	[28,29]
Diosgenin	*Prunus dulcis*, *Trigonella foenum-graecum*, *Dioscorea villosa*, and *Dioscorea japonica*	SW480, DU145, LnCaP, T98G, C6, and A549	47 * µM (A549)	Sustaining Proliferative Signaling, Resisting Cell Death, Activating Invasion and Metastasis, Inducing Angiogenesis, Enabling Replicative Immortality, and Tumor Promoting Inflammation	[30,31,32,33,34,35,36]
Diosgenin-3-O-β-D-fructofuranosyl-(1→6)-[α-L-rhamnopyranosyl-(1→2)]-β-D-glucopyranoside	*Paris polyphylla*	LN229, U251, Capan-2, HeLa, and HepG2	4.18 * µM (LN229), 3.85 * µM (U251), 3.26 * µM (Capan-2), 3.30 * µM (HeLa), and 4.32 * µM (HepG2)	Sustaining Proliferative Signaling and Resisting Cell Death	[37]
Gracillin	*Pairs polyphylla, Dioscorea villosa*, *Aconitum carmichaeli, Solanum incanum,* and *Solanum virginianum*	MDA-MB-231, MCF7, H460, H226B, T47D, MDA-MB-453, and A549	2.421 * μmol/L. (A549)	Deregulating Cellular Energetics and Resisting Cell Death	[38,39]
Paris Saponin I	*Paris polyphylla*	HUVEC and PC-9-ZD	0.643 † µM (HUVEC) and 2.51 * µM (PC-9-ZD)	Resisting Cell Death, Inducing Angiogenesis, Activating Invasion and Metastasis	[40]
Paris Saponin II	*Paris polyphylla*	HUVEC and PC-9-ZD	0.994 † µM (HUVEC) and 3.12 * µM (PC-9-ZD)	Resisting Cell Death, Inducing Angiogenesis, Activating Invasion and Metastasis	[40]
Paris Saponin VI	*Paris polyphylla*	HUVEC and PC-9-ZD	2.204 † µM (HUVEC) and 4.21 * µM (PC-9-ZD)	Resisting Cell Death, Inducing Angiogenesis, Activating Invasion and Metastasis	[40]
Paris Saponin VII	*Paris polyphylla*	MDA-MB-231, MDA-MB-436, MCF-7, HUVEC, and PC-9-ZD	3.16 * µM (MDA-MB-231), 3.45 * µM (MDA-MB-436), 2.86 * µM (MCF-7), 6.212 † µM (HUVEC), and 3.57 * µM (PC-9-ZD)	Resisting Cell Death, Inducing Angiogenesis, Activating Invasion and Metastasis	[40,41]
PP9	*Paris polyphylla*	HT-29 and HCT116	1.08 † µM (HT-29) and 1.97 † µM (HCT116)	Resisting Cell Death, Evading Growth Suppressors	[42]
Protodioscin	*Dioscorea villosa*, *Trigonella foenum-graecum*, and *Asparagus officinalis*	5637 and T24 cells	72.6 * µM (5637) and 63.4 * µM (T24)	Resisting Cell Death, Activating Invasion and Metastasis	[43]
Schizocapsa Plantaginea Hance I (SSPH I)	*Tacca plantaginea*	HepG2	3.395 * µM	Resisting Cell Death	[44]
Taccaoside A	*Tacca plantaginea* and *Tacca subflabellata*	H1299, MHCC97H, BT549, SW620, and HUVEC	-	Avoiding Immune Destruction	[45]
Timosaponin AIII	*Anemarrhena asphodeloides*	H1266 and A549	1.55 † µM (H1266) and 2.16 † µM (A549)	Resisting Cell Death, Activating Invasion and Metastasis	[46]
Trillin	*Trillium tschonoskii*	HepG2 and PLC/PRF	-	Resisting Cell Death, Activating Invasion and Metastasis	[47]
Trilliumoside A	*Trillium govanianum*	A549 and SW-620	1.83 † µM (A-549) and 1.85 † µM (SW-620)	Resisting Cell Death, Activating Invasion and Metastasis	[48]
Trilliumoside B	*Trillium govanianum*	A549 and SW-620	1.79 † µM (A-549) and 3.18 † µM (SW-620)	Resisting Cell Death, Activating Invasion and Metastasis	[48]
Polyphyllin II	*Paris polyphylla*	HepG2, BEL7402, T24, and 5637	4.8351 * μM (HepG2), 4.4765 * μM (BEL7402), 4.43 ± 0.08 µg/mL (T24), and 7.87 ± 0.39 µg/mL (5637)	Resisting Cell Death, Activating Invasion and Metastasis	[49,50]
Polyphyllin E	*Paris polyphylla*	SK-OV-3 and OVCAR-3	7.462 * µM (SK-OV-3) and 5.053 * µM (OVCAR-3)	Resisting Cell Death, Activating Invasion and Metastasis	[51]
Cholestanol Glucoside (CG)	*Lasiodiplodia theobromae*	A549, PC3, HepG2, U251, MCF7, and OVCAR3	-	Resisting Cell Death	[52]
Polyphyllin VII	*Paris polyphylla*	A549	0.41 * µM	Resisting Cell Death	[53]
S-20	*Solanum americanum*	K562, K562/ADR, HL-60, and U937	8.35 µM ± 0.57 (K562), 10.74 µM ± 0.92 (K562/ADR), 22.14 µM ± 0.54 (HL-60), and 29.52 µM ± 1.99 (U937)	Evading Growth Suppressors and Resisting Cell Death	[54]
Polyphyllin B	*Paris polyphylla*	NUGC-3, MKN-1, MKN-45, HGC-27, and NUGC-4	1.447 * µM (NUGC-3), 2.734 * µM (MKN-1), 3.378 * µM (MKN-45), 3.318 * µM (HGC-27), and 2.579 * µM (NUGC-4)	Activating Invasion and Metastasis, Resisting Cell Death, and Evading Growth Suppressors	[55]
Polyphyllin III	*Paris polyphylla*	MDA-MB-231, HS-578T, HBL-100, MCF-7, and T47D	7.96 * µM (MDA-MB-231), 2.59 * µM (HS-578T), 7.74 * µM (HBL-100), 3.89 * µM (MCF-7), and 9.85 * µM (T47D)	Resisting Cell Death	[56]
Polyphyllin D	*Paris polyphylla*	IMR-32, LA-N-2, and NB-69	25 * µM (IMR-32), 20 * µM (LA-N-2), and 5 * µM (NB-69)	Resisting Cell Death	[57]
Polyphyllin I	*Paris polyphylla*	SGC7901/DDP, SGC7901, 143-B, HOS, DU145, and PC3	2.48 * μM (SGC7901), 0.93 * μM (SGC7901/DDP), 0.3942 † µM (143-B), 0.8145 † µM (HOS), 1.03 * µM (DU145), and 2.13 * μM (PC3)	Resisting Cell Death, Activating Invasion and Metastasis	[58,59,60]
Pennogenin-3α-L-rhamnopyranosyl-(1→4)-[α-Lrhamno-pyranosyl-(1→2)]-β-D-glucopyranoside (N45)	*Paris vietnamensis*	U251 and U87	3.808 * μg/mL (U251) and 3.39 * (U87) μg/mL	Resisting Cell Death	[61]
Oleandrin	*Nerium oleander*	A549, SW480, HCT116, RKO, A375, GL261, U87MG, MCF7, SK-BR-3, and MDA-MB-231	47 † nM (A375), 6.07 † nM (MCF7), 1.42 † nM (SK-BR-3), and 11.47 † nM (MDA-MB-231)	Resisting Cell Death	[62,63,64,65,66]
Terrestrosin D	*Tribulus terrestris*	HUVEC and PC-3	-	Resisting Cell Death, Evading Growth Suppressors, and Inducing Angiogenesis	[67]

**Table 2 cancers-15-03900-t002:** The anti-cancer effects of steroidal saponins in vivo.

Compound	Source	In Vivo Model	Targeted Hallmark of Cancer	References
Bufalin	*Bufo gargarizans*	Neuroblastoma, Gallbladder Cancer	Resisting Cell Death, Activating Invasion and Metastasis	[37,51]
Dioscin	*Dioscorea zingiberensis* and *Dioscorea nipponica*	Lung Adenocarcinoma and Colorectal Cancer	Tumor Promoting InflammationTumor Microenvironment, Activating Invasion and Metastasis, and Deregulating Cellular Energetics	[55,62]
Diosgenin	*Prunus dulcis*, *Prunus amygdalus*, *Trigonella foenum-graecum*, *Dioscorea villosa*, and *Dioscorea japonica*	Colorectal Cancer and Prostate Cancer	Sustaining Proliferative Signaling, Resisting Cell Death, Activating Invasion and Metastasis, and Resisting Cell Death	[28,29]
Gracillin	*Rhizoma paridis*, *Pairs polyphylla*, *Dioscorea villosa*, *AconitumAcontum carmichaeli*, *Solanum incanum*, *Solanum virginianum*, and *Solanum xanthocarpum*	Breast Cancer and Non-small Cell Lung Cancer	Deregulating Cellular Energetics and Resisting Cell Death	[39,54]
Paris Saponin VII	*Paris polyphylla*	Breast Cancer	Resisting Cell Death	[42,60]
Polyphyllin B	*Paris polyphylla*	Gastric Cancer	Resisting Cell Death	[55]
Polyphyllin I	*Paris polyphylla*	Osteosarcoma, Prostate Cancer, and Gastric Cancer	Resisting Cell Death, Activating Invasion and Metastasis	[58,59,60]
Polyphyllin III	*Paris polyphylla*	Breast Cancer	Resisting Cell Death	[56]
PP9	*Paris polyphylla*	Colorectal Cancer	Resisting Cell Death, and Evading Growth Suppressors	[33]
Protodioscin	*Dioscorea villosa*, *Trigonella foenum-graecum*, and *Asparagus officinalis*	Bladder Cancer	Resisting Cell Death, Activating Invasion and Metastasis	[35]
RCE-4	*Reineckea carnea*	Cervical Cancer	Resisting Cell Death, and Tumor Promoting Inflammation	[68]
Taccaoside A	*Tacca plantaginea* and *Tacca subflabellata*	Non-small Cell Lung Cancer	Avoiding Immune Destruction	[61]
Terrestrosin D	*Tribulus terrestris*	Prostate Cancer	Resisting Cell Death, Evading Growth Suppressors, and Inducing Angiogenesis	[67]
Timosaponin AIII	*Anemarrhena asphodeloides*	Non-small Cell Lung Cancer	Resisting Cell Death, Activating Invasion and Metastasis	[46]
Oleandrin	*Nerium oleander*	Glioma	Resisting Cell Death	[66]

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
