# Peer review of "Steroidal Saponins: Naturally Occurring Compounds as Inhibitors of the Hallmarks of Cancer"

_cancers, 2023, doi:10.3390/cancers15153900_

Round 1

Reviewer 1 Report

With the current manuscript, Amin and colleagues cover a series of naturally-occurring steroidal saponins on their potential utility to be developed as anticancer agents. The content of the manuscript fits within the scope of the journal, its conceptualization being also coherent. However, the scientific relevance is partially reduced due to a series of recent manuscripts covering the same and/or a very similar topic (doi: 10.3390/pathophysiology28020017 ; 10.2174/1568026622666220330011047 ). On the other hand, and while the article might constitute a good piece of work to readers, a few issues should be stressed out:

A) The ‘method’ underlying the articles’ selection is briefly described in lines 87-92, but it would be conveniently to further detail as some articles appear to have been missed by the authors. A few examples that were missed include the:

1. effects of oleandrin in breast cancer cells, namely the pro-apoptotic effects mediated through the activation endoplasmic reticulum stress (doi: 10.1016/j.biopha.2020.109852) and the activation of immunogenic cell death via the PERK/elF2α/ATF4/CHOP pathway (doi: 10.1038/s41419-021-03605-y).

2. polyphyllin VII inhibition of caspase-activated DNase by attenuating the PI3K/Akt and NF-κB signaling pathways in A549 human lung cancer cells (doi: 10.3892/mmr.2019.10879).

B) Oftentimes, authors appear to include personal opinions/overviews rather than scientifically-based information. For example, authors claim that natural products have a low toxicity (Lines 61-64) but it is well known that most naturally-occurring compounds do not feed directly the clinical pipeline due to their toxicity and require molecule-to-lead optimization.

C) The taxonomic designation of several species requires revision, as in:

Table 1: Prunus amygdalus is not currently recognized, the correct taxonomic designation being Prunus dulcis.

Rhizoma paridis is not a taxonomic designation, rather Paris polyphylla.

Acontum carmichaeli requires revision to Aconitum carmichaeli.

Solanum xanthocarpum is not currently used, rather the taxonomic synonym Solanum virgianum.

In this matter, authors are advised to check the taxonomic designation on the Plant List (http://www.theplantlist.org/).

D) As in similar reviews, it would be more than convenient to include the chemical structure of the highlighted steroidal saponins, thus enabling a more fruitful analysis from readers working on drug design.

E) Finally, one would expect the authors to highlight (in the conclusions section) those steroidal saponins that have a higher potential to feed investigational pipelines (advanced preclinical and clinical) on anticancer drug development. Such discussion should consider i) the chemical novelty of some steroidal saponins, ii) those with stronger effects than those observed with conventional anticancer drugs, iii) steroidal saponins that display different modes of action or a pleiotropic mechanism.

In this matter, due to the general toxicity of bufadienolides, it is highly unlikely that bufalin might proceed to clinical development (Lines 184-190). Also, diosgenin has a limited structural novelty and has been into the spotlight for several decades, thus being also unlikely to constitute a clinical candidate in cancer therapy.

Author Response

Reviewer 1

  1. A) The ‘method’ underlying the articles’ selection is briefly described in lines 87-92, but it would be conveniently to further detail as some articles appear to have been missed by the authors. A few examples that were missed include the:
  2. effects of oleandrin in breast cancer cells, namely the pro-apoptotic effects mediated through the activation endoplasmic reticulum stress (doi: 10.1016/j.biopha.2020.109852) and the activation of immunogenic cell death via the PERK/elF2α/ATF4/CHOP pathway (doi: 10.1038/s41419-021-03605-y).

Response: Oleandrin is a triterpenoid saponin whereas the focus of this review is steroidal saponin.

  1. polyphyllin VII inhibition of caspase-activated DNase by attenuating the PI3K/Akt and NF-κB signaling pathways in A549 human lung cancer cells (doi: 10.3892/mmr.2019.10879).

Response: Done and yellow highlighted

B) Oftentimes, authors appear to include personal opinions/overviews rather than scientifically-based information. For example, authors claim that natural products have a low toxicity (Lines 61-64) but it is well known that most naturally-occurring compounds do not feed directly the clinical pipeline due to their toxicity and require molecule to-lead optimization.

Response: Done and changes are highlighted.

C) The taxonomic designation of several species requires revision, as in: Table 1: Prunus amygdalus is not currently recognized, the correct taxonomic designation being Prunus dulcis. Rhizoma paridis is not a taxonomic designation, rather Paris polyphylla. Acontum carmichaeli requires revision to Aconitum carmichaeli. Solanum xanthocarpum is not currently used, rather the taxonomic synonym Solanum virgianum. In this matter, authors are advised to check the taxonomic designation on the Plant List (http://www.theplantlist.org/).

Response: Done and changes are highlighted.

D) As in similar reviews, it would be more than convenient to include the chemical structure of the highlighted steroidal saponins, thus enabling a more fruitful analysis from readers working on drug design.

Response: Done in new Figure 2.

E) Finally, one would expect the authors to highlight (in the conclusions section) those steroidal saponins that have a higher potential to feed investigational pipelines (advanced preclinical and clinical) on anticancer drug development. Such discussion should consider

i) the chemical novelty of some steroidal saponins,

ii) those with stronger effects than those observed with conventional anticancer drugs,

iii) steroidal saponins that display different modes of action or a pleiotropic mechanism. In this matter, due to the general toxicity of bufadienolides, it is highly unlikely that bufalin might proceed to clinical development (Lines 184- 190). Also, diosgenin has a limited structural novelty and has been into the spotlight for several decades, thus being also unlikely to constitute a clinical candidate in cancer therapy.

Response: Done and changes are highlighted.

Reviewer 2 Report

The purpose of the manuscript appears well identified: a review of steroidal saponins effects on the hallmarks of cancer, with the aim of providing a foundation for further research. The pharmacological actions of saponins so far investigated, paragraphs 2-11, are sufficiently described, but overall the paper does not meet the quality standards of Cancers.

In more detail:

-Simple summary and abstract look almost identical. Importantly, the literature search strategy and main results must be summarized in the abstract.

-Lines 27-29. In practice, the authors argue that cancer hallmarks targeting with conventional cancer drugs failed to provide ideal results because of toxic side effects, and that naturally derived compouds could be the solution. This statement is pretty superficial, considering that many conventional cancer drugs are naturally derived: it should be appropriately reformulated to introduce the low-toxicity approach.

-Lines 49-50. Ref 2-4 should be amended: (i) Un update of cancer hallmarks is in Cancer Discov 2022;12:31–46, (ii) ref 4 does not refer to an additional hallmark.

-Lines 54-61. Ref 6-9 are not appropriate: they refer to specific tumor histotypes and clinical conditions (6-8) or to specific drug (9). Ref 11 and 15 too are not appropriate in support of general statements.

-The addition of a figure of saponin structure would help understandind the text.

-line 87 (and graphical abstract). Is Tumor microenvironment a cancer hallmark?

Table 1. the in vitro anticancer effects of saponins is shown in this table. The cell list is not easy to read: some are cancer cell lines, some are normal cells, and a legend is needed.  Without an appropriate legend, it is not clear what the IC50 value refers to, and what is its relationship to the targeted cancer hallmark. In the absence of the experimental metodology (biological endpoint, treatment time, etc.) the IC50 has no meaning.

- Conclusions and future directions. This section, of particular relevance in a review, is practically missing.

English language is of good quality

Author Response

Reviewer 2

1- Simple summary and abstract look almost identical. Importantly, the literature search strategy and main results must be summarized in the abstract.

Response: Done and changes are marked.

2 -Lines 27-29. In practice, the authors argue that cancer hallmarks targeting with conventional cancer drugs failed to provide ideal results because of toxic side effects, and that naturally derived compounds could be the solution. This statement is pretty superficial, considering that many conventional cancer drugs are naturally derived: it should be appropriately reformulated to introduce the low-toxicity approach.

Response: Done

3-Lines 49-50. Ref 2-4 should be amended: (i) Un update of cancer hallmarks is in Cancer Discov 2022;12:31–46, (ii) ref 4 does not refer to an additional hallmark.

Response: Ref 2-4 were checked and Ref 4 was modified (Douglas H. Hallmarks of Cancer: New Dimensions. Cancer Discov 2022; 12:31–46).

4-Lines 54-61. Ref 6-9 are not appropriate: they refer to specific tumor histotypes and clinical conditions (6-8) or to specific drug (9). Ref 11 and 15 too are not appropriate in support of general statements.

-The addition of a figure of saponin structure would help understanding the text.

Response:  Ref 6-9 and 11 and 15 are reviewed. Additional figure of steroidal saponin structures is added as Figure 2.

5-line 87 (and graphical abstract). Is Tumor microenvironment a cancer hallmark?

Response: Indeed, targeted therapy may address the cancer microenvironment (Ref: Naser, R., Fakhoury, I., El-Fouani, A., Abi-Habib, R., & El-Sibai, M. (2023). Role of the tumor microenvironment in cancer hallmarks and targeted therapy (Review). International journal of oncology62(2), 23. https://doi.org/10.3892/ijo.2022.5471).

Table 1. the in vitro anticancer effects of saponins is shown in this table. The cell list is not easy to read: some are cancer cell lines, some are normal cells, and a legend is needed. Without an appropriate legend, it is not clear what the IC50 value refers to, and what is its relationship to the targeted cancer hallmark. In the absence of the experimental methodology (biological endpoint, treatment time, etc.) the IC50 has no meaning.

Response: Table 1 has been modified.

6- Conclusions and future directions. This section, of particular relevance in a review, is practically missing.

Response: Done and changes are highlighted.

Reviewer 3 Report

In this review manuscript, Bouabdallah et al. discussed the use of steroidal saponins as inhibitors of the hallmarks of cancer. Cancer is a global health burden, and despite advances in treatment, its incidence and mortality rates continue to rise. Recent studies indicate that steroidal saponins exhibit potent anti-cancer properties with low toxicity. This review provides an overview of their role as inhibitors of cancer hallmarks. This is a useful review topic for scholars in the field. However, some suggestions may help further refine the current article format.

Major comments

1. The studies covered in this review include those published between January 2020 and January 2024. The recommendation should be based on 10 years.

2.  Figure 2 may not be easy to understand the mechanisms of “steroidal saponins induce tumor cell death”. A more specific figure might be required.

3. A paragraph should be included to explain the limitations and current challenges associated with using steroidal saponins as an anticancer treatment. These issues have hindered the successful progression of these drugs into clinical trials and their demonstration of efficacy.

Minor editing of English language required

Author Response

Reviewer 3

  1. The studies covered in this review include those published between January 2020 and January 2024. The recommendation should be based on 10 years.

Response: Done and changes are marked.

  1. Figure 2 may not be easy to understand the mechanisms of “steroidal saponins induce tumor cell death”. A more specific figure might be required.

Response: Figure 2 has been modified and is now Figure 3.

  1. A paragraph should be included to explain the limitations and current challenges associated with using steroidal saponins as an anticancer treatment. These issues have hindered the successful progression of these drugs into clinical trials and their demonstration of efficacy.

Response: Done and changes are highlighted.

Round 2

Reviewer 1 Report

Authors now provide an improved version of the manuscript cancers-2471007, considering some of the issues that were stressed out by the reviewers. Despite the significant improvement, there are still minor issues that are worth to revise.

A. In response to the authors’ comment, oleandrin displays a structural backbone corresponding to a steroid structure.

B. Authors still deliver general and personal overviews, without a clear scientific support, that might be often considered superficial as in:

Lines 28-29: It is erroneous to mention that conventional cancer drugs impose ‘detrimental side effects’, as if they are devoid of clinical efficacy. Indeed, even drug development relying on natural products leads to side effects, as in the case of taxol and vincristine.

Lines 32-34: Following the same rationale of the comment above, authors cannot claim that steroidal saponins possess ‘generally low toxicity’. In fact, several steroidal saponins such as digitalin or bufadienolide exert severe cardiotoxic effects.

Lines 540-542: Authors cannot argue that ‘steroidal saponins’ can be considered ‘effective candidates for the treatment of cancer’. Indeed, and as reviewed by the authors, there are nearly no advanced preclinical data (e.g., in animal models or mechanistic studies) neither are clinical candidates enabling to conclude on this.

C. While revising some of the taxonomic designations, several species still require modification as in:
Table 1: Revise “Solanum virgianum” to “Solanum virginianum”.

Schizocapsa plantaginea is not the correct taxonomic designation, rather the taxonomic synonym Tacca plantaginea.

Solanum nigrum should be replaced by Solanum americanum (Both in table 1 and line 281).

D. As previously advised, authors now include the structures of several steroidal saponins. However, the structure style is not the same, authors being requested to use a chemical drawing tool such as ChemDraw or MNova.

Author Response

Reviewer 1 - Round 2

Authors now provide an improved version of the manuscript cancers-2471007, considering some of the issues that were stressed out by the reviewers. Despite the significant improvement, there are still minor issues that are worth to revise.

  1. In response to the authors’ comment, oleandrin displays a structural backbone corresponding to a steroid structure.

Response: Oleandrin is now included as a steroidal saponin that is discussed in this manuscript (page 13, lines 254-267).

  1. Authors still deliver general and personal overviews, without a clear scientific support, that might be often considered superficial as in:

Lines 28-29: It is erroneous to mention that conventional cancer drugs impose ‘detrimental side effects’, as if they are devoid of clinical efficacy. Indeed, even drug development relying on natural products leads to side effects, as in the case of taxol and vincristine.

Response: This has now been modified to “These conventional cancer drugs have shown significant therapeutic efficacy but continue to impose unfavorable side effects on patients.”  This emphasizes that conventional cancer drugs are indeed very effective but involve side effects that may limit their efficacy (page 1, lines 26-27). This is further discussed with the appropriate references on page 3, lines 63-69.

Lines 32-34: Following the same rationale of the comment above, authors cannot claim that steroidal saponins possess ‘generally low toxicity’. In fact, several steroidal saponins such as digitalin or bufadienolide exert severe cardiotoxic effects.

Response: This has been removed (page 1, line 31). The toxicity of steroidal saponins is now addressed in the conclusions section (page 17, lines 545-554).  

Lines 540-542: Authors cannot argue that ‘steroidal saponins’ can be considered ‘effective candidates for the treatment of cancer’. Indeed, and as reviewed by the authors, there are nearly no advanced preclinical data (e.g., in animal models or mechanistic studies) neither are clinical candidates enabling to conclude on this.

Response: This has been modified in the conclusions section (page 17, lines 533-534).

  1. While revising some of the taxonomic designations, several species still require modification as in:
    Table 1: Revise “Solanum virgianum” to “Solanum virginianum”.

Schizocapsa plantagine is not the correct taxonomic designation, rather the taxonomic synonym Tacca plantaginea.

Solanum nigrum should be replaced by Solanum americanum (Both in table 1 and line 281).

Response: This has been modified and highlighted for your convenience.

  1. As previously advised, authors now include the structures of several steroidal saponins. However, the structure style is not the same, authors being requested to use a chemical drawing tool such as ChemDraw or MNova.

Response: A new figure representing the chemical structure of steroidal saponins has been added. ChemDraw was used to construct this figure.

Reviewer 2 Report

Overall, the authors responded adequately to reviewers' comments. The article was significantly improved and is ready for publication.

minor editing is required

Author Response

Reviewer 2 - Round 2

We are very thankful for your constructive comments that helped improve of this manuscript.

Reviewer 3 Report

Accept in present form.

Author Response

Responses to Reviewer - Round 2

We are very thankful for your constructive comments that helped improve this manuscript.

Round 3

Reviewer 1 Report

 Authors now provide an improved version of the manuscript, nearly suitable for publication in Cancers.

Just very minor suggestions before the final acceptance:

A. Line 18: There are steroidal saponins sourced also from animals and microorganisms. Correct “…plant-derived compounds…” to “…naturally-occurring compounds…”.

B. There is no description on Table 1.

C. Please italicize “Bufo gargarizans” (Tables 1 and 2).

D. Please correct the taxonomic designation Anemarrhena Asphodeloides (Tables 1 and 2) as the specific epithet (i.e., second part of the name) is written in lowercase.

E. There is no need to refer that structures were built in ChemDraw as a caption of Figure 2.

Author Response

Response to Reviewer 1 (Round 3)

Line 18: There are steroidal saponins sourced also from animals and microorganisms. Correct “…plant-derived compounds…” to “…naturally-occurring compounds…”.

Response: The suggested comment has been addressed and properly highlighted.

There is no description on Table 1.

Response: Description of Table 1 is now highlighted.

Please italicize “Bufo gargarizans” (Tables 1 and 2).

Response: The suggested comment has been addressed and properly highlighted.

Please correct the taxonomic designation Anemarrhena Asphodeloides (Tables 1 and 2) as the specific epithet (i.e., second part of the name) is written in lowercase.

Response: The suggested comment has been addressed and properly highlighted.

There is no need to refer that structures were built in ChemDraw as a caption of Figure 2.

Response: “ChemDraw” is now removed from the caption of Figure 2.